# Schizophrenia-Like Behavioral Impairments in Mice with Suppressed Expression of Piccolo in the Medial Prefrontal Cortex

**DOI:** 10.3390/jpm11070607

**Published:** 2021-06-26

**Authors:** Atsumi Nitta, Naotaka Izuo, Kohei Hamatani, Ryo Inagaki, Yuka Kusui, Kequan Fu, Takashi Asano, Youta Torii, Chikako Habuchi, Hirotaka Sekiguchi, Shuji Iritani, Shin-ichi Muramatsu, Norio Ozaki, Yoshiaki Miyamoto

**Affiliations:** 1Department of Pharmaceutical Therapy and Neuropharmacology, Faculty of Pharmaceutical Sciences, Graduate School of Pharmaceutical Sciences, University of Toyama, Toyama 930-0194, Japan; ntk3izuo@pha.u-toyama.ac.jp (N.I.); kohei-1572@kem.biglobe.ne.jp (K.H.); ryo.inagaki.a2@tohoku.ac.jp (R.I.); d2062302@ems.u-toyama.ac.jp (Y.K.); fukequan6962@hotmail.com (K.F.); tasano@pha.u-toyama.ac.jp (T.A.); miyamoto@josai.ac.jp (Y.M.); 2Jiangsu Key Laboratory of New Drug Research and Clinical Pharmacy, Xuzhou Medical University, Xuzhou 221004, China; 3Department of Psychiatry, Graduate School of Medicine, Nagoya University, Nagoya 466-8550, Japan; youtat@med.nagoya-u.ac.jp (Y.T.); akariha1029@yahoo.co.jp (C.H.); sguchi77@gmail.com (H.S.); iritani@med.nagoya-u.ac.jp (S.I.); ozaki-n@med.nagoya-u.ac.jp (N.O.); 4Open Innovation Center, Division of Neurological Gene Therapy, Jichi Medical University, Shimotsuke 329-0498, Japan; muramats@jichi.ac.jp; 5Center for Gene and Cell Therapy, The Institute of Medical Science, The University of Tokyo, Tokyo 108-8639, Japan

**Keywords:** piccolo, presynaptic cytomatrix protein, medial prefrontal cortex, dorsal striatum, schizophrenia, optogenetics

## Abstract

Piccolo, a presynaptic cytomatrix protein, plays a role in synaptic vesicle trafficking in the presynaptic active zone. Certain single-nucleotide polymorphisms of the Piccolo-encoding gene *PCLO* are reported to be associated with mental disorders. However, a few studies have evaluated the relationship between Piccolo dysfunction and psychotic symptoms. Therefore, we investigated the neurophysiological and behavioral phenotypes in mice with Piccolo suppression in the medial prefrontal cortex (mPFC). Downregulation of Piccolo in the mPFC reduced regional synaptic proteins, accompanied with electrophysiological impairments. The Piccolo-suppressed mice showed an enhanced locomotor activity, impaired auditory prepulse inhibition, and cognitive dysfunction. These abnormal behaviors were partially ameliorated by the antipsychotic drug risperidone. Piccolo-suppressed mice received mild social defeat stress showed additional behavioral despair. Furthermore, the responses of these mice to extracellular glutamate and dopamine levels induced by the optical activation of mPFC projection in the dorsal striatum (dSTR) were inhibited. Similarly, the Piccolo-suppressed mice showed decreased depolarization-evoked glutamate and -aminobutyric acid elevations and increased depolarization-evoked dopamine elevation in the dSTR. These suggest that Piccolo regulates neurotransmission at the synaptic terminal of the projection site. Reduced neuronal connectivity in the mPFC-dSTR pathway via suppression of Piccolo in the mPFC may induce behavioral impairments observed in schizophrenia.

## 1. Introduction

Psychiatric disorders appear as abnormal behavioral and mental patterns that cause severe distress or disability to the individual. Schizophrenia, one of serious psychiatric disorders, is characterized by three major symptoms, positive and negative symptom and cognitive deficit, as well as an accompanying neurological phenomenon [1,2,3]. Current clinical treatments for schizophrenia have been proven effective for positive symptoms, with only a few being effective for negative symptoms and cognitive deficit. Therefore, elucidating pathophysiological conditions related to schizophrenia is imperative for the early development of highly effective therapeutic treatments.

Many genetic and environmental risk factors are commonly involved in schizophrenia, and the overlap between such risk factors plays a potential role in its development. In addition, some parts of the brain such as the frontal cortex, amygdala, hippocampus, and thalamus are involved in schizophrenia, and the disconnection between such parts in the brain, rather than impairment in one part of the brain, results in its symptoms. Especially, many patients with schizophrenia exhibit functional, structural, and metabolic abnormalities in the prefrontal cortex (PFC) in neuroimaging studies reporting on the effects of antipsychotic medications and cognitive remediation therapy [4]. The dopaminergic dysfunction in the PFC and high density of the binding of the antipsychotic drug chlorpromazine in the PFC in schizophrenic patients support this notion [5]. D2 receptors are typically distributed in the subcortical regions, such as the striatum; however, functional brain imaging studies have shown that the density of striatal D2 receptors is altered in schizophrenia patients without medication [6]. Therefore, it is thought that altered neuronal activity and connection in the PFC by the genetic and environmental insults is associated with the pathophysiology of schizophrenia [7,8,9].

Piccolo, a presynaptic protein encoded by the *PCLO* gene, plays a role in synaptic vesicle trafficking by interacting with several proteins in the presynaptic active zone [10,11,12,13,14]. Ultrastructurally, the presynaptic active zone is an electron-dense, largely detergent-resistant cytomatrix comprising multiple scaffolding proteins, including Munc18, SNAP-25, and Piccolo [15,16], hypothesized to function as synaptic vesicle regulators in neurotransmission process due to their interaction [17]. Piccolo has been reported to be involved in cognitive function, synaptic plasticity, and psychostimulant-induced psychosis [18,19]. A genome-wide association study showed that single-nucleotide polymorphisms (SNPs) in the *PCLO* gene were significantly associated with bipolar disorder and major depressive disorder. Indeed, the SNP rs13438494 in intron 24 of *PCLO*, which disturbs the splicing pattern of PCLO mRNA to decrease expression [20], is associated with bipolar disorder [21]. Symptoms of drug dependence-related parameters, such as the age of first exposure to a psychostimulant, tobacco dependence, and fentanyl requirement for pain relief in human, have been associated with Piccolo SNPs [22]. In addition, SNP rs2522833 in exon 19 of *PCLO*, causing amino acid substitution (from serine to alanine) in the Ca^2+^ binding C2A domain of Piccolo, induce the mild increase of synaptic transmission [23]. Its C allele is the top risk variant of major depressive disorder [24] and also associated with the reduced regional brain volume [25,26], lower memory performance [27], and increased activity in the left amygdala during processing of fearful faces [28]. There is no consensus of the clinical association between the functional status of Piccolo and mental disorders. Piccolo physiologically expresses at the terminals of glutamatergic and GABAergic neurons [11]. Since Piccolo alteration in its function or expression could change the balance of circuit activity, the phenotypic effects to the brain circuits and behavior are necessary to be investigated.

Therefore, in the present study, we investigate whether or not the suppressed expression of the *Pclo* gene in the medial PFC (mPFC) of mice using adeno-associated virus (AAV) *Pclo* miRNA vectors affects the neurophysiological function. We also investigated whether or not a causal relationship between Piccolo dysfunction and behavioral impairments in mental disorders could be established.

## 2. Materials and Methods

### 2.1. Animals

Seven-week-old C57BL/6J male mice (Nihon SLC, Hamamatsu, Japan) were used for this study to avoid the effects of the estrous cycle. The weights of the mice ranged from 21–26 g. The number of mice used was described in each experiment. The mice were housed in a 12-h light–dark cycle (lights on at 8:00) with food and water available ad libitum.

All procedures were performed in accordance with the National Institutes of Health Guide for the Care and Use of Laboratory Animals and Guidelines for the Care and Use of Laboratory Animals at the University of Toyama (Approval No. A2018PHA-5).

### 2.2. Drugs

Risperidone (Sigma-Aldrich, St. Luis, MO, USA, R3030, 0.01 mg/kg) dissolved with saline was administered by intraperitoneal (i.p.) injections 30 min before each behavioral test. Especially, administration was performed on each day of the novel object recognition and fear-conditioning test. Saline was injected to the control mice as a vehicle. Bicuculine and NBQX were purchased from Tocris Bioscience (Bristol, UK).

### 2.3. Production and Microinjection of AAV Vector

This study used an AAV vector production protocol described in earlier studies [29,30]. Viral vectors (AAV1) were designed to express an antisense sequence for *Pclo* (TGCTGATCCCAAACTGTCACCTCCAAGTTTTGGCCACTGACTGACTTGG AGGTCAGTTTGGGAT) and an enhanced GFP sequence (AAV- *Pclo* miRNA/EGFP vectors) based on murine miR-155 as a backbone (BLOCK-iT; Invitrogen Japan K493500, Thermo Fisher Scientific, Tokyo, Japan) under cytomegalovirus promotor. Viral vectors containing only the enhanced GFP sequence (AAV-EGFP vectors) were used as controls. All procedures were performed in accordance with the Guideline for Recombinant DNA Experiment by the Ministry of Education, Culture, Sports, Science and Technology, Japan and were approved by the Gene Recombination Experiment Safety Committee at the University of Toyama (Approval No. G2015PHA-14).

The microinjection of AAV vectors (0.7 μL/side, 0.1 μL/min) into the mPFC of the mice (+1.6 mm anterior, ±0.3 mm lateral from the bregma, and +1.6 mm ventral from the skull) according to the brain atlas [31] was performed as described in a previous report [32]. The mice were used for experiments four weeks after the AAV-injections. Four weeks after AAV injection, behavioral experiments were conducted in the following order: locomotor activity test, spatial working test, novel object recognition test, and spatial working test. After the serial experiments, samples from these mice were used for immunostaining, Western blotting, or quantitative real-time polymerase chain reaction (PCR). Another group of mice was used for electrophysiological recordings or in vivo microdialysis four weeks after the AAV-injection.

### 2.4. Quantitative Real-Time PCR

All reactions were performed in duplicate using the following cycling protocol: enzyme heat activation for 10 min at 95 °C, 40 cycles of denaturation at 95 °C for 30 s, annealing at 59 °C for 40 s, and extension at 72 °C for 60 s. Piccolo primers used for real-time PCR were as follows: 5′-TGCCTGGTTTCTTCTCAGATGT-3′ (forward: 753–774 base pairs [bp]) and 5′-GAGTCTGATATCAAATCAAAAGGGT-3′ (reverse: 816–840 bp), 5′-GTCAAAACAGCCAGCAGTCC-3′ (forward: 14,607–14,626 bp) and 5′-GTCCATGAGATCGGAGATGG-3′ (reverse: 14,752–14,771 bp). A 36B4 transcript quantified using the forward primer 5′-ACCCTGAAGTGCTCGACATC-3′ and reverse primer 5′-AGGAAGGCCTTGACCTTTTC-3′ was used as the internal control.

### 2.5. Western Blotting

Western blotting was performed using standard methods [33] with the following primary antibodies: anti-Piccolo (abcam ab20664, Cambridge, UK; 1:1000), anti-phospho-synapsin I (Phosphosolutions 1560-6267, Aurora, CA, USA; 1:1000), anti-synapsin I (Enzo Life Sciences BML-SA495, Farmingdale, NY, USA; 1:1000), anti-synaptophysin (Sigma-Aldrich S5768; 1:1000), anti-munc18 (BD Biosciences 610336, San Jose, CA, USA; 1:1000), anti-SNAP-25 (BD Biosciences 610366; 1:1000), and anti-α-Tubulin (Santa Cruz Biotechnology sc-5546, Santa Cruz, CA, USA; 1:1000). Proteins were detected using horseradish peroxidase-conjugated secondary antibodies (GE Healthcare, Amersham, UK; NA9310, 1:5000) and the ECL Prime kit (GE Healthcare, RPN2332).

### 2.6. Electrophysiological Recordings

A 64-channel multi-electrode dish system (Alpha MED Sciences, Tokyo, Japan) was employed, as described in an earlier study [18]. We used MED-P515A probes (Alpha MED Sciences) with a 150-μM interpolar distance as electrodes, a chamber depth of 10 mm, and 64 planar microelectrodes in an 8 × 8 array. For paired pulse facilitation, two field excitatory postsynaptic potentials (fEPSPs) were evoked with twin pulses at interpulse intervals of 20, 60, and 100 ms. The ratio of the second versus the first potential was determined. Long-term potentiation (LTP) was elicited by theta burst stimulation (TBS; 20 trains, each train of 4 pulses at 100 Hz, intertrain interval of 200 ms, total train duration of 40 ms). After the TBS stimulation, fEPSPs were recorded for 60 min at 1-min intervals per slice.

### 2.7. In Vivo Microdialysis

In vivo microdialysis was performed as described in a previous study [34,35]. The guide cannula was placed into the mPFC (+1.6 mm anterior, 0.3 mm lateral from the bregma, and +1.6 mm ventral from the skull) or dSTR (1.2 mm anterior, 1.0 mm lateral from the bregma and 2.7 mm ventral from the skull) (26). Dialysate was collected at a flow rate of 0.5 or 1.0 μL/min for dopamine (DA) or glutamate (Glu) in 10- or 15-min fractions, respectively, and injected into the high-performance liquid chromatography (HPLC) system (HTEC-500; Eicom, Kyoto, Japan). To measure GABA, dialysate was collected at a flow rate of 0.5 μL/min in 30-min fractions using a fraction collector and mixed with a cocktail of ortho-phthalaldehyde (4 mmol/L) and carboxylic acid solutions (pH 9.5) containing 0.04% mercaptoethanol for 1.5 min. The mixed dialysate solution was continuously autoinjected into the HPLC system.

### 2.8. Optogenetic Stimulation

For optogenetic stimulation, AAV- *Pclo* miRNA/EGFP or AAV-EGFP vectors mixed with AAV9 vectors expressing ChIEF under the human synapsin promoter were microinjected into the mPFC (+1.7 mm anterior, 0.3 mm lateral from the bregma, and +1.5 mm ventral from the skull) (Paxinos and Franklin, 2008). ChIEF is a light-sensitive protein that is generated by introducing some mutagenesis to the original channelrhodopsin 2 (ChR2), and its activation in response to the light resembles more natural spiking patterns than ChR2 [36]. Four weeks after the microinjection, guide cannula (AGFL-4, Eicom) for an optical fiber and in vivo microdialysis probe (A-FL-4-01, Eicom) were implanted in the dSTR (+1.2 mm anterior, 1.0 mm lateral from the bregma and 2.7 mm ventral from the skull). For optogenetic stimulation, the mice received blue light pulses (pulse width, 15 ms; frequency, 10 Hz; intensity, 5 mW, laser, 473 nm) for 15 min (ESFI-700, Eicom) under the control of stimulation scheduler (ESST-800, Eicom).

### 2.9. Behavioral Analyses

Locomotor activity test

To measure the locomotor activity in a novel environment, a mouse was placed in a transparent acrylic cage (45 × 45 × 40 cm), and its locomotion was measured every 5 min for a total duration of 60 min using a Scanet MV-40 (Melquest, Toyama, Japan).

PPI test

An SR-LAB system (San Diego, CA, USA) was used to measure the startle response and prepulse inhibition (PPI). The test session began by placing a mouse in a plastic cylinder and leaving it undisturbed for 10 min. The background noise level in each chamber was 70 dB. The intensity of the startle stimulus was 120 dB. The prepulse sound (74 or 78, 82, and 86 dB) was presented 100 ms before the startle stimulus. Four combinations of prepulse and startle stimuli were used (74 and 120, 78 and 120, 82 and 120, and 86 and 120 dB). The extreme outliers determined by Smirnov–Grubbs test were removed.

Novel object recognition test

Mice were individually habituated to an open-field box (30 × 30 × 45 cm) for 3 days. During the acquisition phase, two objects were placed symmetrically in the center of the chamber for 10 min, and the mouse’s exploratory behavior was analyzed. Twenty-four hours later, one object was replaced by a novel object, and the exploratory behavior of the mouse was analyzed again for 10 min. Discrimination of novel object was assessed by counting the total time and number of contacts.

Contextual learning test

In the contextual learning test, the mice were trained and tested for two consecutive days. The test was performed in operant chambers. Contextual fear conditioning was measured using the FCC mode system (Melquest, Toyama, Japan). Training involved allowing the subjects to freely explore the operant chamber for 2 min. Thereafter, electrical foot shocks (0.6 mA, five seconds) with an auditory cue were automatically delivered through a grid floor using customized programming four times with 15-second intervals. The mice were returned to their home cages after completing the conditioning procedure. Twenty-four hours after training, the mice were brought back to the same chamber, and freezing behavior was observed and recorded for 4 min.

Y-maze test (spontaneous alternation behaviors)

The Y-maze test was conducted using an apparatus with three identical arms. In this test, a mouse was placed at the end of one arm of the apparatus and allowed to move freely during an 8-min session. An alternation was defined as entry into all three arms on consecutive choices. The percentage of spontaneous alternation behavior was calculated as follows: [(Alternations)/(Total number of arms entered − 2)] × 100.

Mild social defeat stress exposure

Before the social interaction and forced swimming tests, the mice were subjected to subthreshold levels of social defeat stress. This consisted of three consecutive 5-min defeat sessions in a single day, with 15 min of rest between each session. During this time, the mice were directly exposed to an aggressive male ICR mouse (>8 weeks old; Nihon SLC). The social interaction test was performed 48 h after the exposure and subsequently followed by the forced swimming test 96 h after the exposure.

Social interaction test

The social interaction test comprised two 150-s phases, separated by a duration of 30 s. During the first phase, a stress-exposed mouse was placed in the open-field arena (45 × 45 × 40 cm) of an empty wire-mesh enclosure (8 × 10 × 8 cm). During the second phase, the mouse was placed in the open-field arena, with an ICR mouse present in the wire-mesh enclosure. The social interaction time, i.e., the duration in which the subject mouse stayed within the interaction zone (14 × 27 cm around the wire-mesh enclosure) was measured. During the 30-s break between each phase, the subject mouse was transferred back to its home cage. The ICR mouse used for the test had no previous interaction with the subject mouse.

Forced swimming test

The forced swimming test was performed as previously described in a previous study [32]. In brief, the mice were placed in a transparent polycarbonate cylinder (21 cm in diameter × 22.5 cm high) containing water at 22 °C at a depth of 18 cm. The mice were then forced to swim for 6 min. The duration of immobility was measured in 1-min as a period using a SCANET MV-40AQ (Melquest).

### 2.10. Statistical Analyses

All data are expressed as the mean ± standard error of the mean. Statistical differences between the two groups were determined using Student’s *t*-test. Statistical analyses related to the behavioral test was re-performed by one-way ANOVA and *post-hoc* multiple comparisons using Tukey’s test. In the microdialysis analysis, statistical differences were determined by a repeated measures two-way ANOVA followed by Bonferroni’s *post-hoc* test when *F* ratios were significant (*p* < 0.05). All statistical analyses were performed by Prism version 5 (Graph Pad Software, San Diego, CA, USA).

## 3. Results

### 3.1. Mice with a Suppressed Piccolo Expression in the mPFC

First, mice with a suppressed Piccolo expression in the mPFC (miPiccolo mice) were produced via AAV- *Pclo* miRNA/EGFP vector microinjection (Figure 1A, upper panel). Immunohistochemical studies showed an obvious EGFP expression in the mPFC of miPiccolo mice (Figure 1A, lower panel). The *Pclo* mRNA expression in the mPFC of miPiccolo mice was downregulated to about half that of control Mock mice (Student’s *t*-test, n = 6, *Pclo* (753–840): *p* = 0.0001, *Pclo* (14607–14771): *p* = 0.0401; Figure 1B). The Piccolo protein expression in the miPiccolo mice was also suppressed in the mPFC (Student’s *t*-test, n = 6, *p* = 0.0122; Figure 1C). The miPiccolo mice also showed a decreased expression of presynaptic protein SNAP-25 in the mPFC, while the Munc18 and Synaptophysin levels remained unchanged (Student’s *t*-test, n = 6, SNAP-25: *p* = 0.0007, Munc18: *p* = 0.4016, Synaptophysin: *p* = 0.6339; Figure 1D). Furthermore, the degree of phosphorylation at the CaMKII regulatory site in Synapsin I (Ser603) was lower in miPiccolo than in Mock mice (Student’s *t*-test, n = 6, *p* = 0.0038; Figure 1E).

### 3.2. Diminished Synaptic Properties in the mPFC by Piccolo Suppression

To examine the synaptic properties of mPFC neurons, electrophysiological recordings of mPFC slices from miPiccolo mice were conducted. Analyses of paired-pulse facilitation, an indicator of presynaptic functions, revealed that miPiccolo mice had significantly lower ratios at an interstimulus interval of 20 and 60 ms than Mock mice (Student’s *t*-test, n = 5, Interval 20: *p* = 0.0331, Interval 60: *p* = 0.0073, Interval 100: *p* = 0.7503; Figure 2A).

Next, LTP was examined in the mPFC. The suppression of Piccolo diminished LTP induction (average of 55–60 min, Student’s *t*-test, n = 6, *p* = 0.0055; Figure 2B, right panel). Repeated theta-burst stimulation was then used to determine the LTP saturation, which indicated the functional retention of synaptic modification. The Mock mice had greater LTP saturation than the miPiccolo mice (average of 55–60 min, Student’s *t*-test, n = 6, *p* = 0.0001; Figure 2C, right panel), suggesting that presynaptic dysfunction diminished the synaptic plasticity in the mPFC of miPiccolo mice.

### 3.3. Reduction in Depolarization-Evoked Glutamate and Dopamine Elevation in the mPFC by Piccolo Suppression

To clarify the changes in neurotransmission, the high-potassium ion (K^+^)-induced elevation of extracellular neurotransmitter levels in the mPFC of miPiccolo mice was assessed using an in vivo microdialysis method. Accordingly, miPiccolo mice were found to have significantly lower basal levels of extracellular Glu in the mPFC than Mock mice (Student’s *t*-test, n = 6, *p* = 0.0300; Figure 3A, left panel). In contrast, there was no marked difference in the basal levels of extracellular DA between miPiccolo and Mock mice (Student’s *t*-test, n = 5, *p* = 0.8845; Figure 3B, left panel). An increase in the extracellular neurotransmitter levels by high-K^+^-induced (100 mM) depolarization was observed in both Mock and miPiccolo mice (Figure 3A,B, right panel). However, there was a significant reduction in the depolarization-evoked Glu and DA elevation in the mPFC of miPiccolo mice (ANOVA with repeated measurement, Glu: n = 6, *F*_(9,90)_  =  4.232, *p* = 0.0001; Figure 3A, right panel and DA: n = 5, *F*_(8,64)_  =  6.701, *p* < 0.0001; Figure 3B, right panel).

### 3.4. Diminished Neuronal Responses in the mPFC-dSTR Pathway by Piccolo Suppression

Changes to the neuronal function in the mPFC might significantly influence other brain regions. We therefore examined the brain region projecting from the mPFC by injecting an AAV-EGFP vector to the mPFC in order to visualize neuronal terminus with GFP fluorescence, indicating the projection from mPFC to dSTR (Figure 3C, left panel).

To investigate the neuronal function in the mPFC-dSTR pathway, changes in the extracellular Glu and DA levels in the dSTR were measured in response to optogenetic activation. Optogenetic stimulation of mPFC-dSTR projection was achieved by the local injection of an AAV vector expressing the light-sensitive protein CHIEF and laser stimulation in the dSTR (Figure 3C, right panel). The basal level of Glu without optical stimulation was significantly decreased in the dSTR of miPiccolo mice compared with CHIEF-AAV injection group, while that of DA was not markedly changed (Student’s *t*-test, Glu: n = 6, *p* = 0.0014; Figure 3D, left panel and DA: n = 6, *p* = 0.6458; Figure 3E, left panel). Optical stimulation in the dSTR induced an increase in extracellular Glu and a decrease in extracellular DA levels through photosensory protein expression in the mPFC of Mock mice via the AAV-ChIEF vector injection (Figure 3D,E, right panel). However, responses to the optical stimulation of mPFC projections in the dSTR of miPiccolo mice were diminished (ANOVA with repeated measurement, Glu; n = 6, *F*_(14,140)_  =  3.517, *p* < 0.0001; Figure 3D, right panel and DA: n = 6; *F*_(11,110)_  =  5.943, *p* < 0.0001; Figure 3E, right panel). These results suggest that Piccolo induces abnormal behavior via the regulation of mPFC-dSTR projection in the extracellular Glu and DA levels in the dSTR.

To determine whether decrease of DA in dSTR under optical stimulation is mediated by glutamatergic transmission, NBQX, an antagonist for AMPA type of glutamate receptors, was applied to the microdialysis. The application of NBQX (20 μM) without optical stimulation reduced the extracellular DA in the dSTR (Appendix A). In comparison, NBQX application with optical stimulation induced a transient increase in the extracellular DA in the dSTR, which seems to be a relatively slight change compared to the condition without NBQX (Appendix A). These data suggest that blockade of glutamatergic transmission decreases the extracellular DA increase in the dSTR under optical stimulation.

### 3.5. Enhanced Depolarization-Evoked Dopamine Elevation Via Disinhibition of GABAergic Regulation in the dSTR by Piccolo Suppression

The high-K^+^-induced elevation in extracellular neurotransmitters in the dSTR was measured to confirm the diminished connections in the mPFC-dSTR pathway in miPiccolo mice. miPiccolo mice showed significantly lower basal levels of extracellular Glu in the dSTR than Mock mice (Student’s *t*-test, n = 6, *p* = 0.0135; Figure 3F, left panel), which is consistent with the results obtained in Figure 3D, left panel. In contrast, there was no observable differences in the basal DA or GABA levels between Mock and miPiccolo mice (Student’s *t*-test, DA: n = 6, *p* = 0.9375; Figure 3G, left panel and GABA: n = 6, *p* = 0.2835; Figure 3H, left panel). The miPiccolo mice exhibited a greater reduction in depolarization-evoked Glu and GABA elevation in the dSTR than Mock mice (ANOVA with repeated measurement, Glu: n = 6, *F*_(12,120)_ = 5.572, *p* < 0.0001; Figure 3F, right panel and GABA: n = 6; *F*_(5,50)_  =  2.672, *p* = 0.0323; Figure 3H, right panel). Conversely, miPiccolo mice exhibited an enhanced depolarization-evoked DA elevation in the dSTR compared with Mock mice (ANOVA with repeated measurement, n = 6; *F*_(14,140)_  =  2.207, *p* = 0.0102; Figure 3G, right panel). Taken together, these findings indicate that depolarization due to high-K^+^ stimulation induced elevations of extracellular Glu, DA, and GABA levels in the dSTR, and these responses were enhanced in miPiccolo mice.

To investigate the regulation of extracellular DA level by GABAergic transmission, bicuculline, a GABA receptor antagonist, was applied to the microdialysis. Without optical stimulation, bicuculline (50 µM) elevated the extracellular DA levels in the dSTR (Appendix A). The effects of the blockade of the GABAergic transmission with optical stimulation was examined. However, all mice that received the combination of optical stimulation and bicuculline application (10–50 µM) to the dSTR died (n = 3) during or just after the measurement. Mice receiving bicuculline application (20–50 μM) during optical stimulation showed increased extracellular DA levels before death (Appendix A). Collectively, these results suggest blockade of GABAergic transmission elevates the extracellular DA levels in the dSTR.

### 3.6. Disruption of Locomotor Activity, Sensorimotor Gating, and the Cognitive Function by Piccolo Suppression

The miPiccolo mice showed a significantly increased locomotor activity in a novel environment compared to Mock mice (one-way ANOVA followed by Tukey’s *post-hoc* test, n = 12; Mock-Saline vs. miPiccolo-Saline, *p* = 0.0117; Figure 4A, right panel). Significant impairment in the 78-dB prepulse/120-dB pulse trial was observed in miPiccolo mice compared to Mock mice when the acoustic startle response and PPI (psychometric measure of sensorimotor gating) were measured (one-way ANOVA followed by Tukey’s *post-hoc* test, n = 11–12, 78–120 db, Mock-Saline vs. miPiccolo-Saline *p* = 0.0383, Figure 4B).

To examine the role of Piccolo in the mPFC on learning and memory, Mock and miPiccolo mice were subjected to behavioral tests requiring precise cognitive control. First, the novel object recognition test was performed to measure the ability of the mice to recognize a familiar object. The Mock mice showed cognitive preference, as indicated by a longer exploration time for the novel object than for a familiar object. However, the miPiccolo mice showed a significantly shorter exploration time for the novel object than did the Mock mice (one-way ANOVA followed by Tukey’s *post-hoc* test, n = 12, Mock-Saline vs. miPiccolo-Saline, *p* < 0.0001; Figure 4C).

In the spatial context learning test, we found that the miPiccolo mice exhibited significantly less freezing than Mock mice for context-elicited fear following conditioning (one-way ANOVA followed by Tukey’s *post-hoc* test, Mock-Saline vs. miPiccolo-Saline, *p* = 0.0064; Figure 4D).

To determine the performance of immediate spatial working memory, the mice were subjected to the Y-maze test. There was decreased spontaneous alternation in the Y-maze test in the miPiccolo mice (one-way ANOVA followed by Tukey’s *post-hoc* test, Mock-Saline vs. miPiccolo-Saline *p* = 0.0189; Figure 4E). As expected, the miPiccolo mice displayed schizophrenia-like behavioral impairments related to positive symptoms and cognitive deficits. Therefore, we evaluated the effect of risperidone (0.01 mg/kg) in Mock and miPiccolo mice. The miPiccolo mice receiving risperidone exhibited a marked reduction in locomotor activity in a novel environment compared to miPiccolo mice receiving saline (one-way ANOVA followed by Tukey’s *post-hoc* test, n = 12, miPiccolo-Saline vs. miPiccolo- Risperidone *p* = 0.0017; Figure 4A, right panel). In addition, risperidone treatment reversed the PPI impairment induced by Piccolo suppression in the mPFC (one-way ANOVA followed by Tukey’s *post-hoc* test, n = 11–12, 82–120 db, miPicclo-Saline vs. miPiccolo-Risperidone *p* = 0.0332, Figure 4B). In contrast, risperidone treatment did not improve the cognitive deficits in miPiccolo mice at doses effective for alleviating hyperlocomotion and PPI deficit (Figure 4C–E).

### 3.7. Stress Vulnerability Induced by Piccolo Suppression

To determine changes in stress sensitivity induced by Piccolo suppression in the mPFC, the miPiccolo mice were exposed to mild social defeat stress, after which the behavioral tests were performed. There were no marked differences in the interaction time in the social interaction test or the immobility time in the forced swim test between non-stress- and stress-exposed Mock mice (Figure 5A,B). However, stress-exposed miPiccolo mice had a significantly shorter interaction time in the social interaction test than non-stress-exposed miPiccolo mice (one-way ANOVA followed by Tukey’s *post-hoc* test, n = 14, miPiccolo/non-stress vs. miPiccolo/stress *p* = 0.0049; Figure 5A). Similarly, a significant difference in the immobility time in the forced swim test was observed between non-stress- and stress-exposed miPiccolo mice (one-way ANOVA followed by Tukey’s *post-hoc* test, n = 13–14, miPiccolo/non-stress vs. miPiccolo/stress *p* = 0.0003; Figure 5B). Thus, miPiccolo mice are suggested to be vulnerable to stress exposure.

## 4. Discussion

In the CNS, the synaptic proteins in the cytomatrix at the active zone (CAZ) in the presynaptic neuron maintain the structural and functional integrity of the synaptic vesicle pools and recruit L-type voltage-dependent Ca^2+^ channels to the presynaptic membrane. This allows the exocytosis of neurotransmitters from the synaptic vesicles [37]. Piccolo, a CAZ protein, can therefore regulate synaptic strength at the presynaptic neurons. In the present study, we showed that local Piccolo suppression in the mPFC resulted in diminished paired-pulse facilitation and LTP in the same brain region. Supporting this result, we observed reductions in the basal levels of extracellular Glu and depolarization-evoked elevations in extracellular Glu levels, which reflects the presynaptic Glu content, in the mPFC of miPiccolo mice. Our observations in electrophysiological and neurochemical analyses suggest that Piccolo suppression causes deficits in synaptic strength due to reduced neurotransmitter exocytosis from synaptic vesicles in the glutamatergic interneuron of the mPFC, whose impairment is related schizophrenia [38]. Furthermore, we observed changes in levels of CAZ proteins associated with the exocytosis of synaptic vesicles (Synapsin I phosphorylation and SNAP-25 expression) in the mPFC of miPiccolo mice, which supports this notion. Thus, Piccolo modulates the synaptic vesicle reserve pool and homeostasis of CAZ protein in vivo. Previously, Yang et al. generated brain-specific SNAP-25 knockout mice by crossbreeding of CaMKIIα-Cre mice of SNAP-25 flox mice [39]. The resultant mice exhibited schizophrenia-like behaviors accompanied by the reduction of SNAP-25 expression in the cortical and hippocampal regions. However, these mice also show the potent increase of glutamate levels in the cortex, which is opposite to the clinical evidence of schizophrenia, that is the glutamatergic dysfunction. In our present study, suppressed expression of Piccolo decreased the extracellular glutamate levels in the prefrontal cortex, following the clinical consensus. Schizophrenia-like behavioral impairments in prefrontal downregulation of Piccolo are probably not mediated by observed reduction of SNAP-25. Piccolo may play a vital role in the regulation of neurotransmitter release.

In combination experiments of microdialysis and optogenetics, we found that optical stimulation of the mPFC-dSTR projection induced the elevation of Glu and reduction of DA. The application of NBQX in the dSTR inhibited the reduction of the extracellular DA level, suggesting that the activation of mPFC-dSTR projection negatively regulated the extracellular DA level in the dSTR mediated by glutamatergic transmission. However, without optical stimulation, the application of NBQX in the dSTR reduced the DA levels, suggesting that the extracellular DA levels in the dSTR receive positive regulation by the local glutamatergic system. Therefore, the relationship between extracellular DA in the dSTR and its regulation by glutamatergic transmission is complicated. In addition, bicuculline application to the dSTR increased the extracellular levels of DA, which suggests that the DA in the dSTR is under the regulation of GABAergic transmission. This does not contradict the elevation of extracellular DA levels observed with the combination of bicuculline application and optical stimulation, although the mice suffered a fatal outcome. Our notion that extracellular DA levels in the dSTR are regulated by glutamatergic and GABAergic transmission is supported by the findings of previous studies. In the neuronal circuits in the STR, glutamatergic projection neurons from the PFC provide excitatory input to GABA neurons in the STR, and GABA neurons in the STR regulate the DA nerve terminals projected from the substantia nigra [40,41]. However, the DA responses in the dSTR under optical and depolarizing stimulation are completely different. This discrepancy may be due to limitations in the methodology, as optogenetic stimulation of the terminals from the mPFC may increase the D2 autoreceptor activity or induce changes in the cholinergic neurotransmission, and depolarizing stimulation may also exert a wide range of mechanisms that could alter DA release. Further analyses of the neuronal composition of the projection from mPFC to dSTR and the relationship between glutamatergic and GABAergic transmissions are necessary. Indeed, our optical stimulation system cannot excite specific neurons, so the auto receptors of D2 and cholinergic neuronal system could be regulated. Many mechanisms potentially underlying the DA release alternation via optical stimulation and depolarization by high K+ should be considered. However, such experiments involve certain technical limitations at present, so no all phenomenon can be explained by knockdown of Piccolo in the mPFC alone.

It is well-known that the glutamatergic and dopaminergic neuronal systems in the PFC control the cognitive functions. Changes in these functions are closely linked to mental disorders, especially schizophrenia. The glutamatergic neuronal system is known to mediate episodic memory. In schizophrenia patients, the PFC exhibits a reduced glutamatergic neuronal activity and NMDA receptor expression compared with that in healthy persons [42,43]. The dopaminergic neuronal system in the PFC is involved in higher-order executive functions and working memory [44,45,46,47]. Functional abnormalities of DA receptors have been observed in the PFC of schizophrenia patients [48]. In the present study, a behavioral analysis of Piccolo suppression in the mPFC showed disruptions in episodic memory, such as object recognition and spatial context learning, as well as working memory. These disruptions in the cognitive function were not improved by the administration of the atypical antipsychotic drug risperidone. Similarly, clinically used antipsychotic drugs are not particularly effective in improving cognitive dysfunction in schizophrenia patients. Therefore, the reduced glutamatergic and dopaminergic neuronal responses by Piccolo suppression in the PFC are considered to be similar to the pathophysiological conditions seen in schizophrenia patients.

Current clinical treatments for schizophrenia have proven effective for ameliorating positive symptoms, but only a few treatments have been effective against negative symptoms and cognitive deficits. Therefore, elucidating the pathophysiological conditions related to schizophrenia is crucial for the early development of efficient therapeutic treatments. The two-hit hypothesis has been proposed as an etiology of schizophrenia. Many genetic and environmental risk factors are commonly involved in schizophrenia. The overlap between these risk factors plays a potential role in the development of schizophrenia [49]. The genetic suppression of Piccolo in the mPFC showed the following schizophrenia-like behavioral impairments: enhanced locomotor activity as a positive symptom; decreased objective, spatial and working memories as cognitive deficits; and impaired PPI as an accompanying symptom. Such genetic manipulation made the mPFC highly susceptible to environmental stress. In addition, exposure of the mice to mild social defeat stress led to social withdrawal and a diminished motivation similar to the negative symptoms of schizophrenia. Some observed schizophrenia-like behavioral impairments improved after treatment with the atypical antipsychotic drug risperidone. Furthermore, at the molecular and cellular levels, Piccolo suppression in the mPFC led to a reduced expression of the presynaptic protein SNAP-25. These findings are consistent with the observations made in the postmortem PFC of schizophrenia patients [50,51,52]. Some brain regions, such as the PFC, STR, amygdala, hippocampus, and thalamus, are involved in the pathophysiology of schizophrenia. Indeed, neuroimaging studies on the effects of antipsychotic medications and cognitive remediation therapy have shown that many schizophrenia patients exhibit functional, structural, and metabolic abnormalities in the PFC [4]. Therefore, it has been hypothesized that neuronal disconnection between the PFC and other brain regions, rather than abnormalities in a single brain region, is associated with the pathophysiology of schizophrenia [7,8,9]. As mentioned above, functional disconnection was noted in the mPFC-dSTR pathway due to the suppression of Piccolo. Thus, miPiccolo mice had the face and predictive validities for schizophrenia. Piccolo-suppressed mice may be useful animal models for schizophrenia, suggesting a decline in the expression or function of Piccolo in the PFC of schizophrenia patients.

Clinically, SNPs, which reduce the expression of *PCLO*, are related to the higher risks of mental disorders [20,22,24]. In contrast, the piccolo levels of the autopsy brains from patients with schizophrenia are controversial (Appendix A and [53]). This is probably because the most patients diagnosed with schizophrenia received the treatments of antipsychotics, which are reported to increase the expression of Piccolo (Appendix A and [54]). In the present study, mice with the knockdown of *Pclo* gene in the mPFC exhibit similarity in the behavioral patterns. Our findings suggest that Piccolo expression decreased in the patients with schizophrenia under the untreated conditions.

In conclusion, our findings show that the presynaptic cytomatrix protein Piccolo plays a functional role in the regulation of neurotransmitter exocytosis from the CAZ in vivo. Its suppressed expression in the mPFC affects not only the neuronal activity in the mPFC but also the neuronal transmission mediated by Glu, GABA, and DA in the dSTR. Such neuronal dysfunction can cause behavioral impairments that resemble schizophrenia symptoms. Accordingly, the suppressed expression and/or function of Piccolo in the PFC is considered one aspect of the pathophysiological condition in patients with schizophrenia. In addition, these Piccolo-suppressed mice may be useful as a novel animal model for schizophrenia. Further clarification regarding the role of Piccolo in schizophrenia will help offer new insight into psychiatry and the therapeutic treatment of schizophrenia.

## Figures and Tables

**Figure 1 jpm-11-00607-f001:**
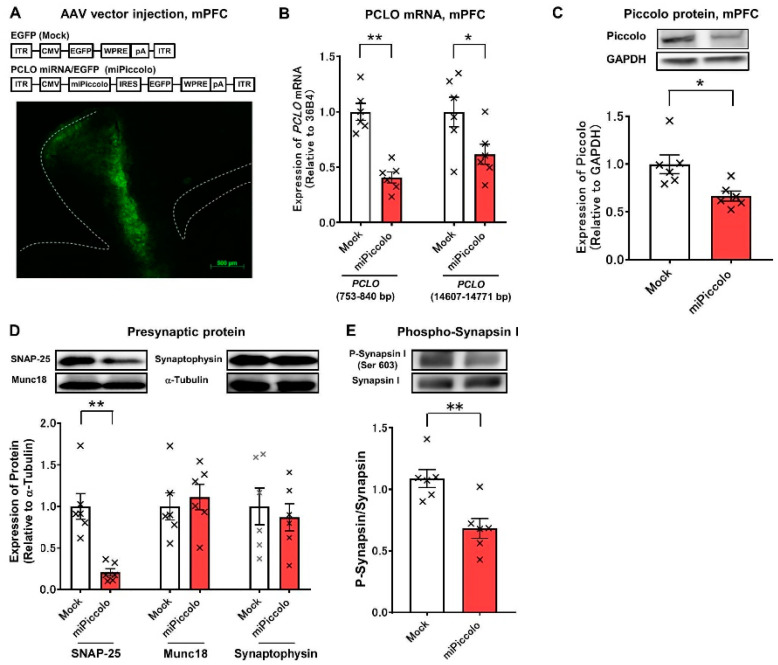
Generation of miPiccolo mice. (**A**) Upper panel, Sequence of AAV-EGFP or AAV- *Pclo* miRNA/EGFP vectors. The AAV vector was constructed using the cytomegalovirus immediate-early promoter (CMV) to drive EGFP or *Pclo* miRNA/EGFP. ITR: inverted terminal repeats, IRES: internal ribosomal entry site, WRPE: woodchuck hepatitis virus post-transcriptional regulatory element, pA: polyadenylation signal sequences. Lower panel, a fluorescent microscopic image of GFP in the mPFC of miPiccolo mouse (sagittal section). (**B**) The expression of *Pclo* mRNA was measured by quantitative real-time RT-PCR and presented as relative to the expression of the housekeeping gene 36B4 (n = 6). * *p* < 0.05, ** *p* < 0.01 vs. Mock (Student’s *t*-test). (**C**) The expression of Piccolo was measured by Western blotting and presented as relative to the expression of GAPDH (n = 6). * *p* < 0.05 vs. Mock (Student’s *t*-test). (**D**) The expression of presynaptic proteins in the mPFC. The bar graphs show the quantification of the expression of each protein compared with the expression of α-tubulin (n = 6). ** *p* < 0.01 vs. Mock (Student-*t* test). (**E**) The phosphorylation of Synapsin I in the mPFC was measured by Western blotting and presented as relative to the expression of Synapsin I (n = 6). ** *p* < 0.01 vs. Mock (Student’s *t*-test).

**Figure 2 jpm-11-00607-f002:**
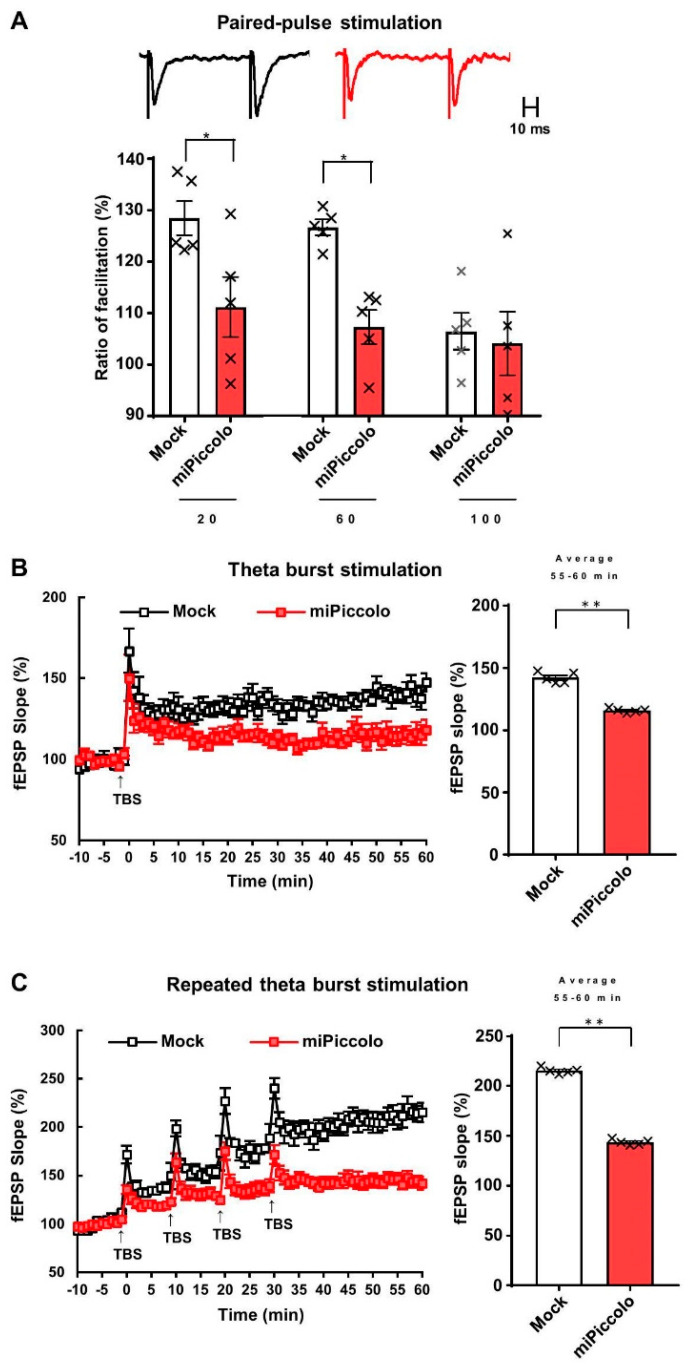
An electrophysiological analysis in the mPFC of miPiccolo mice. (**A**) Ratio of paired-pulse facilitation in the mPFC (n = 5). Representative slope pairs evoked with 60-ms interstimulus intervals are shown for the brain slice acutely prepared from the Mock and miPiccolo mice. * *p* < 0.05 vs. Mock (Student’s *t*-test). (**B**) Left panel: Long-term potentiation (LTP) in the mPFC (n = 5). fEPSP: field excitatory postsynaptic potential, TBS: theta burst stimulation (TBS). Right panel: The potentiation rate was calculated by comparing the average slope 55–60 min after the TBS (n = 5). ** *p* < 0.01 vs. Mock (Student’s *t*-test). (**C**) Left panel: Saturation of LTP in the mPFC (n = 5). Right panel: The potentiation rate was calculated by comparing the average slope 55–60 min after the first TBS (n = 5). ** *p* < 0.01 vs. Mock (Student’s *t*-test).

**Figure 3 jpm-11-00607-f003:**
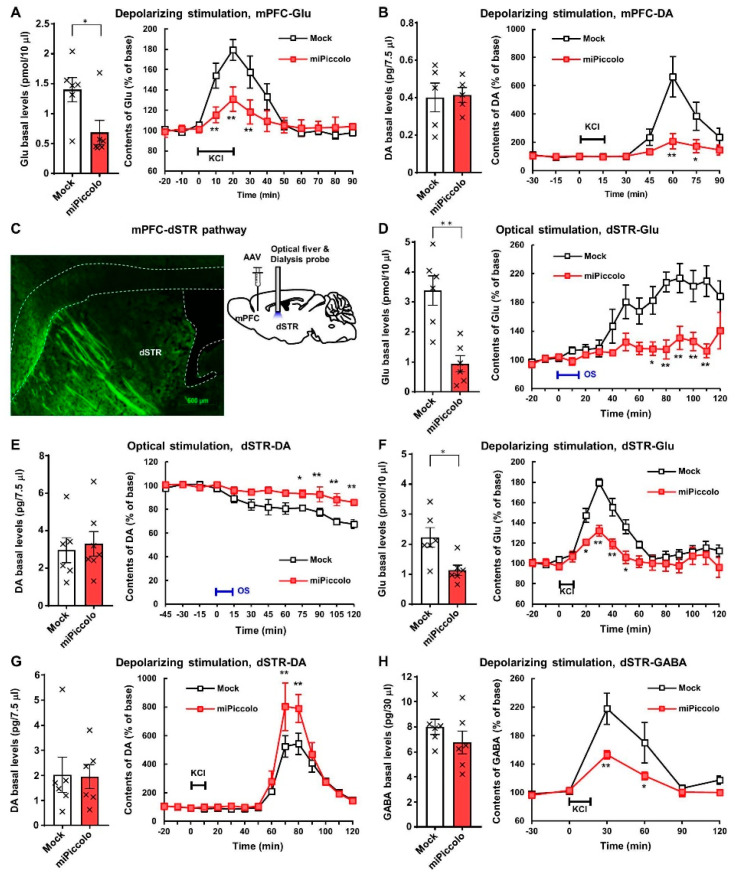
In vivo brain microdialysis in the mPFC and dSTR of miPiccolo mice. (**A**) Left panel: Basal extracellular levels of glutamate (Glu) in the mPFC (n = 6). * *p* < 0.05 vs. Mock (Student’s *t*-test). Right panel: High K^+^-stimulated (100 mM, 20 min) Glu elevation in the mPFC (n = 6). ** *p* < 0.01 vs. Mock (ANOVA with repeated measurement followed by Bonferroni’s *post-hoc* test). (**B**) Left panel: Basal extracellular levels of dopamine (DA) in the mPFC (n = 5). Right panel: High-K^+^-stimulated (100 mM, 15 min) DA elevation in the mPFC (n = 5). * *p* < 0.05, ** *p* < 0.01 vs. Mock (ANOVA with repeated measurement followed by Bonferroni’s *post-hoc* test). (**C**) Left panel: A fluorescence microscopic image of GFP in the dSTR of miPiccolo mouse (sagittal section). Right panel: An image of the AAV vector injection site in the mPFC and the optical fiber and dialysis showing the cannulation site in the dSTR of the mouse. (**D**) Left panel: Basal extracellular levels of Glu in the dSTR (n = 6). ** *p* < 0.01 vs. Mock (Student’s *t*-test). Right panel: Change in the Glu levels by optogenetic stimulation (15 min) in the dSTR (n = 6). * *p* < 0.05, ** *p* < 0.01 vs. Mock (ANOVA with repeated measurement followed by Bonferroni’s *post-hoc* test). (**E**) Left panel: Basal extracellular levels of DA in the dSTR (n = 6). Right panel: Change in the DA levels by optogenetic stimulation (15 min) in the dSTR (n = 6). * *p* < 0.05, ** *p* < 0.01 vs. Mock (ANOVA with repeated measurement followed by Bonferroni’s *post-hoc* test). (**F**) Left panel: Basal extracellular levels of Glu in the dSTR (n = 6). * *p* < 0.05 vs. Mock (Student’s *t*-test). Right panel: High-K^+^-stimulated (100 mM, 10 min) Glu elevation in the dSTR (n = 6). * *p* < 0.05, ** *p* < 0.01 vs. Mock (ANOVA with repeated measurement followed by Bonferroni’s *post-hoc* test). (**G**) Left panel: Basal extracellular levels of DA in the dSTR (n = 6). Right panel: High-K^+^-stimulated (100 mM, 10 min) DA elevation in the dSTR (n = 6). ** *p* < 0.01 vs. Mock (ANOVA with repeated measurement followed by Bonferroni’s *post-hoc* test). (**H**) Left panel: Basal extracellular levels of GABA in the dSTR (n = 6). Right panel: High-K^+^-stimulated (100 mM, 10 min) GABA elevation in the dSTR (n = 6). * *p* < 0.05, ** *p* < 0.01 vs. Mock (ANOVA with repeated measurement followed by Bonferroni’s *post-hoc* test).

**Figure 4 jpm-11-00607-f004:**
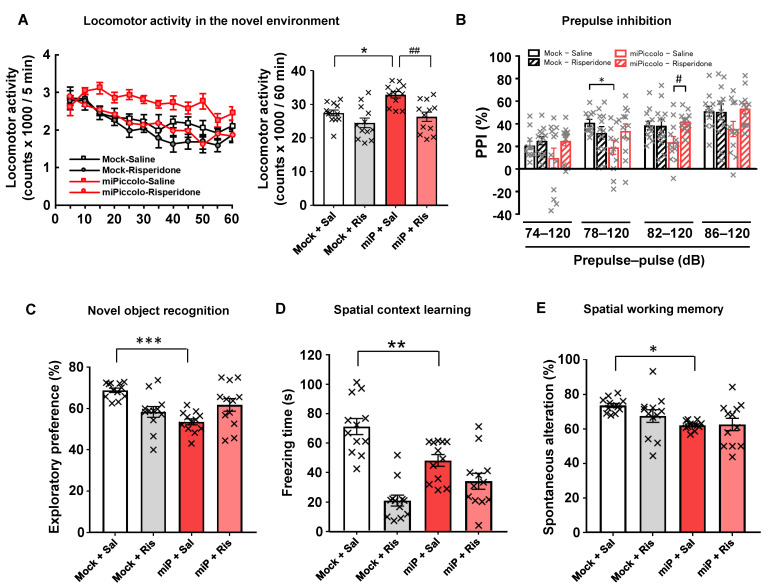
Behavioral analyses of miPiccolo mice. (**A**) The locomotor activity in a novel environment was measured every 5 min for 60 min (n = 12). * *p* < 0.05 vs. Mock-Saline, ^##^
*p* < 0.01 vs. miPiccolo-Saline (one-way ANOVA followed by Tukey’s *post-hoc* test). (**B**) Prepulse inhibition (PPI) was measured by presenting a semi-random series of prepulse of various intensities (74, 78, 82, 86 decibels) paired with the acoustic startle stimulus (120 db) to the mice (n = 12). * *p* < 0.05 vs. Mock-Saline, ^#^
*p* < 0.05 vs. miPiccolo-Saline (one-way ANOVA followed by Tukey’s *post-hoc* test). (**C**) An indication of the time spent approaching the novel object in the trial phase of the novel object recognition test. Risperidone was administered 30 min before the acquisition and test phase (n = 12). *** *p* < 0.001 vs. Mock (one-way ANOVA followed by Tukey’s *post-hoc* test). (**D**) Spatial context learning was assessed based on the freezing time at 24 h after a conditioning session. Risperidone was administered 30 min before the training and test (n = 12). ** *p* < 0.01 vs. Mock (one-way ANOVA followed by Tukey’s *post-hoc* test). (**E**) Spatial working memory was assessed based on spontaneous alterations in the Y-maze test (n = 12). * *p* < 0.05 vs. Mock (one-way ANOVA followed by Tukey’s *post-hoc* test). Risperidone (0.01 mg/kg, i.p) was administered 30 min before the behavioral test.

**Figure 5 jpm-11-00607-f005:**
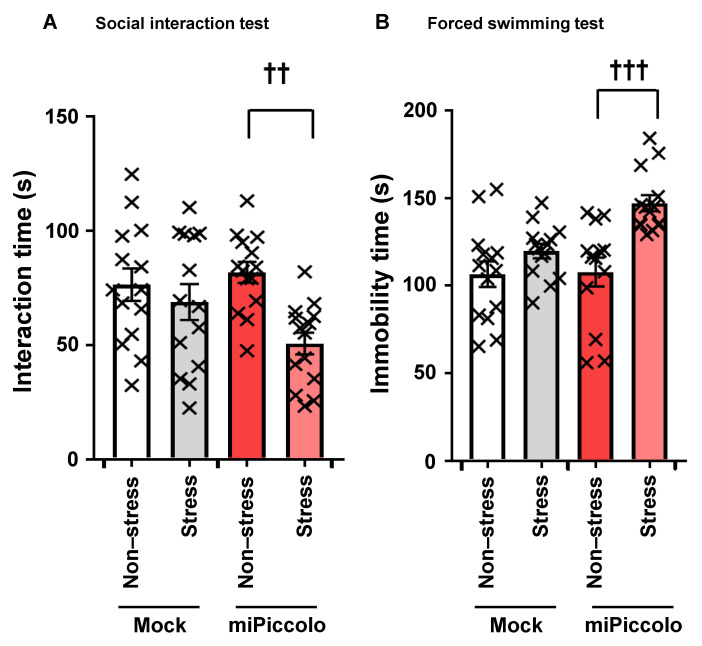
Stress sensitivity in miPiccolo mice. (**A**) After exposure to mild social defeat stress, the time in the interaction zone was measured for 10 min (n = 14). ^††^
*p* < 0.01 vs. miPiccolo/non-stress (one-way ANOVA followed by Tukey’s *post-hoc* test). (**B**) After exposure to mild social defeat stress, the immobility time in water was measured for the final 5 min during a 6-min time frame (n = 13–14). ^†††^
*p* < 0.001 vs. miPiccolo/non-stress (one-way ANOVA followed by Tukey’s *post-hoc* test).

## Data Availability

Data presented in this manuscript are available upon request from the corresponding authors on reasonable request.

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
