# Peer review of "Schizophrenia-Like Behavioral Impairments in Mice with Suppressed Expression of Piccolo in the Medial Prefrontal Cortex"

_jpm, 2021, doi:10.3390/jpm11070607_

Round 1

Reviewer 1 Report

Authors investigated the neurophysiological and behavioral phenotypes in mice with Piccolo suppression in the medial prefrontal cortex (mPFC). Very interesting results and well written manuscript. The manuscript also raised several questions:

Major:

  • All the single-nucleotide polymorphisms of the Piccolo-encoding gene PCLO result in decreased expression of piccolo protein? If so, direct state that the single-nucleotide polymorphisms of the Piccolo-encoding gene PCLO resulting in decreased expression of the protein will lead reader to grasp the reason of this study carried out in the mice with decreased piccolo-expression. If not, how could authors interpret and generalize the results in the case of SNP not decreased expression and functions of piccolo. Need clarification and discussion.
  • In Figure 1C, piccolo-suppressed mice show decreased expression of SNAP-25 in mPFC. What are the phenotypes of SNAP-25 deficient or suppressed mice in measurements shown in the manuscript? Is there overlap of these two proteins in neuron function? If so, the phenotypes of piccolo-suppressed mice might be resulted from the decreased expression of SNAP-25. Need to be explained and thoroughly discussed.
  • Figure 3:
    • Add reference for the measurement of an in vivo microdialysis method.
    • Captions of Left panels in Figure 3 are too small to read.
    • Do the extracellular contents on the left panels, such as Glu in Figure 3A, have any effect on the responses measured in Right panels? Need to be clarified.

Minor:

  • Typos, such as corroborates in line 452.

Author Response

Reply to Reviewer #1’ s Comments

We appreciate your constructive comments. The response were prepared to each comments. The related descriptions and references were added in the revised manuscript.

<Major>

  • All the single-nucleotide polymorphisms of the Piccolo-encoding gene PCLO result in decreased expression of piccolo protein? If so, direct state that the single-nucleotide polymorphisms of the Piccolo-encoding gene PCLO resulting in decreased expression of the protein will lead reader to grasp the reason of this study carried out in the mice with decreased piccolo-expression. If not, how could authors interpret and generalize the results in the case of SNP not decreased expression and functions of piccolo. Need clarification and discussion.

SNP rs13438494 in intron 24 of PCLO, which disturbs the splicing pattern of PCLO mRNA to decrease expression, is associated with bipolar disorder. On the other hand, SNP rs2522833 in exon 19 of PCLO, which induce the mild increase of synaptic transmission, is associated with major depressive disorder and also with the reduced regional brain volume, lower memory performance, and increased activity in the left amygdala during processing of fearful faces. Thus, there is currently no consensus of the clinical association between the functional status of Piccolo and mental disorders. Piccolo physiologically expresses at the terminals of glutamatergic, and GABAergic neurons. Piccolo alteration in its function or expression could change the balance of circuit activity. What will totally in the brain circuits and behavior is necessary to be investigated. The comment and reference were added in the revised manuscript (line 85-87, 90-99, and  ref) [21, 24, 26-29]).

  • In Figure 1C, piccolo-suppressed mice show decreased expression of SNAP-25 in mPFC. What are the phenotypes of SNAP-25 deficient or suppressed mice in measurements shown in the manuscript? Is there overlap of these two proteins in neuron function? If so, the phenotypes of piccolo-suppressed mice might be resulted from the decreased expression of SNAP-25. Need to be explained and thoroughly discussed.

Previously, Yang et al. generated brain-specific SNAP-25 knockout mice by crossbreeding of CaMKIIa-Cre mice of SNAP-25 flox mice (Neural Plast. 2017; 2017: 4526417. doi: 10.1155/2017/4526417.). The resultant mice exhibited schizophrenia-like behaviors accompanied by the reduction of SNAP-25 expression in the cortical and hippocampal regions. However, these mice also show the potent increase of glutamate levels in the cortex, which is opposite to the clinical evidence of schizophrenia, that is the glutamatergic dysfunction. In our present study, suppressed expression of Piccolo decreased the extracellular glutamate levels in the mPFC, following the clinical consensus. Schizophrenia-like behavioral impairments in prefrontal downregulation of Piccolo is probably not mediated by observed reduction of SNAP-25. Related comment and reference were added to the revised manuscript (line 457-466 and ref [40]).

  • Figure 3:
    • Add reference for the measurement of an in vivo microdialysis method.
    • Captions of Left panels in Figure 3 are too small to read.
    • Do the extracellular contents on the left panels, such as Glu in Figure 3A, have any effect on the responses measured in Right panels? Need to be clarified.

The reference with a description of the in vivo dialysis method was added in the revised manuscript (line 178 and ref [35]). The captions of the left panels in Figure 3 were enlarged. The same modification was performed in Fig. 2. As described the second paragraph in the discussion part, the reductions in the baseline extracellular levels of Glu (left graph) and KCl-evoked Glu release, reflecting presynaptic content of Glu, could be caused by the dysfunction of synaptic vesicle reserve pool induced by Piccolo downregulation. The related comment was added in the revised manuscript (line 449).

<Minor>

  • Typos, such as corroborates in line 452.

The word “corroborates” was replaced to “supported” (line 455).

Reviewer 2 Report

The manuscript entitled, “Schizophrenia-like Behavioral Impairments Due to Neuronal Disconnection in Mice with Suppressed Expression of Piccolo in the Medial Prefrontal Cortex” describes how virally mediated downregulation of the presynaptic protein Piccolo in the mouse mPFC leads to neurochemical changes in glutamate, dopamine and GABAergic neurotransmission the mPFC and dorsal striatum as well as behavioral abnormalities modeling some aspects of schizophrenia.

In the mPFC, reduced mPFC Piccolo levels lead to 1) reduced levels of SNAP-25 and phosphorylated Synapsin I, 2) alterations in paired-pulse stimulation and reduced long-term potentiation, and 3) reduced basal glutamate and reduced glutamate and dopamine elevations evoked by local depolarization (by high potassium).

In the dorsal striatum, reduced mPFC Piccolo levels lead to 1) reduced basal glutamate and reduced elevations in glutamate induced by optogenetic stimulation or depolarization, but 2) attenuated reduction of dopamine induced by local optogenetic stimulation, and 3) even enhanced dopamine elevation evoked by depolarization. Also 4) reduced GABA elevation evoked by depolarization was observed.

At behavioral level, the downregulation of mPFC Piccolo leads to novelty induced hyperactivity, disrupted prepulse inhibition, disrupted novel object recognition 24 h later, reduced fear freezing in context associated with foot-shocks and reduced spatial working memory. Also, social interaction was reduced and immobility time increased in the forced swim test in the mice harboring downregulated mPFC Piccolo and subjected earlier to social-defeat stress. Acute treatment with the antipsychotic risperidone reversed the novelty-induced hyperactivity and disrupted prepulse inhibition of these mice. Novel object recognition, fear-freezing in context or spatial working memory defects were not ameliorated by risperidone, but it should be noted that risperidone appeared to worsen these performances also in the control mice. Based on these results the authors conclude that the downregulated Piccolo in the mouse mPFC models positive, negative and cognitive symptoms of schizophrenia.

The results are of interest, experiments appear to be properly performed and the conclusions drawn from the results appear reasonable. However, the authors should themselves carefully once more proof read the whole manuscript and correct inconsistencies.

I have several concerns regarding the manuscript. These should be carefully addressed.

Firstly, it was not clear in what type of neurons Piccolo protein was downregulated in the mPFC. What is known about virally mediated miRNA, which neurons are targeted, how widely distributed? This should be covered in more detail. Secondly, if this protein is presynaptic, should it be downregulated also in the projection sites such as the dorsal striatum and many other well-known projection sites of mPFC projection neurons. Were protein levels analyzed in other brain regions, e.g. striatum, in addition to the mPFC?

Optogenetic methodology was very shortly described. Please, describe in detail the virus construct (source if commercially obtained). Was the light-cable cannulation (give details) inserted at the same time and together with the microdialysis probe, was this performed at the same time as virus injections? In what type of cells and at what subcellular locations are the virally mediated light-sensitive protein expected to be expressed? What is this protein? How was its expression verified in the neurites projected from the mPFC to the dorsal striatum? In Fig. 3C, after which virus is this GFP expression shown in the dorsal striatum? Comparing this representative image 3C with Fig.1A, why any projections are not observed in Fig.1A image? In Fig. 1A, the GFP expression appears very locally restricted. Could you provide images in both cases where both mPFC and striatum are seen simultaneously? Is the virus construct containing GFP the same in Fig. 1A and 3C?

Title is somewhat misleading, in my opinion, it is not shown in this manuscript that the neuronal disconnection causes the observed behavioral impairments. Also, the abbreviated title is not appropriate (observed impairments may resemble some aspects of schizophrenia, but not convincingly “psychosis”). Please rephrase both titles.

Abstract. Results shown in Fig. 1 and 2 are not mentioned in the abstract at all.

Representative images of parvalbumin immunohistochemistry (Fig. 1F) appear to demonstrate that there are even more positive cells in the miPiccolo image. Indicate which cells are counted as positive in these images. Give a micrometer scale for these images and detailed location within the mPFC.

Discussion contains unnecessary repetition. It is recommended that the authors rewrite the discussion to make it more compact.

Other comments

Line 66, unusualness. Please change this.

Line 75, damages, would “insults” fit better here?

Line 82: given references do not directly link Piccolo to psychostimulant-induced psychosis, please specify this

Line 110, in fear-conditioning and novel object recognition tests, did you administer risperidone on both days, please specify this.

Line 216: discrimination of novel object

Lines 223-224: how many foot-shocks given during which time, was freezing analyzed immediately after, did it differ?

Lines 128-132: Please, list all behavioral tests here. Spatial working test mentioned two times, why? If pcr, wester blot and immunohistochemistry were performed after behavioral experiments, how did you control possible confounding effects of behavioral tests e.g. fear-conditioning, risperidone treatments, social-defeat, forced-swim test?

Lines 148-153: What was detection method for parvalbumin immunohistochemistry, did you use perfused and fixed brains, section thickness and plane; sagittal, coronal or horizontal? Give more details about quantification, which parts of mPFC included?

Lines 309-310: p=0.8 but in Fig.3B left panel marked as significant.

Lines 324-326: injected where, what kind of protein is CHIEF? Where is it expressed? Any reference?

Line 331: If this photosensory protein is expressed in the mPFC, how does optical stimulation in the striatum leads to stimulation?

Line 339: antagonist for AMPA type of glutamate receptors

Line 363: miPiccolo enhanced depolarization evoked DA levels in the striatum (Fig. 3G)

Line 377-379 and all other results with two-way ANOVA: Indicate always also the factor (treatment, drug, interaction) when giving the results of two-way ANOVA. Now, it is said that there is a significant increase…, but F and p=0.134?

Line 396: Did you analyze freezing immediately after foot-shocks, did it differ? Did you analyze cue-induced freezing in a different context? Did it differ?

Line 403: p=0.222? Line 409: p=0.134

Lines 423 and 427: give F and p values for each factor (treatment, stress) and interaction

Line 449: Are there glutamatergic interneurons in the mPFC? How downregulated Piccolo affects glutamatergic pyramidal neurons sending collaterals to other pyramidal neurons and GABAergic interneurons? What is known about Piccolo in cortical GABAergic interneurons?

Line 469: medium spiny GABAergic neurons?

Line 494: remove some

Line 521: disconnection

Lines 528-536: unnecessary repetition

Line 533: In Fig 4B, you show that risperidone improved PPI in miPiccolo at least at 82dB prepulse, did two-way ANOVA support this observation? Please, add two-way ANOVA results and comment of this in Results section.

Lines 539-541: What is the policy of this journal in reporting unpublished findings?

Line 542: Is your method knocking down gene or its expression?

Fig. 1D: check y-axis

Fig. 2: figure legend, please give plane of slices used

Fig. 4: figure legend, please mention when risperidone was given in the fear conditioning and object recognition tests, 1st or 2nd day?

Fig. 5B: check y-axis

Author Response

Reply to Reviewer #2’ s Comments

The manuscript entitled, “Schizophrenia-like Behavioral Impairments Due to Neuronal Disconnection in Mice with Suppressed Expression of Piccolo in the Medial Prefrontal Cortex” describes how virally mediated downregulation of the presynaptic protein Piccolo in the mouse mPFC leads to neurochemical changes in glutamate, dopamine and GABAergic neurotransmission the mPFC and dorsal striatum as well as behavioral abnormalities modeling some aspects of schizophrenia. In the mPFC, reduced mPFC Piccolo levels lead to 1) reduced levels of SNAP-25 and phosphorylated Synapsin I, 2) alterations in paired-pulse stimulation and reduced long-term potentiation, and 3) reduced basal glutamate and reduced glutamate and dopamine elevations evoked by local depolarization (by high potassium). In the dorsal striatum, reduced mPFC Piccolo levels lead to 1) reduced basal glutamate and reduced elevations in glutamate induced by optogenetic stimulation or depolarization, but 2) attenuated reduction of dopamine induced by local optogenetic stimulation, and 3) even enhanced dopamine elevation evoked by depolarization. Also 4) reduced GABA elevation evoked by depolarization was observed. At behavioral level, the downregulation of mPFC Piccolo leads to novelty induced hyperactivity, disrupted prepulse inhibition, disrupted novel object recognition 24 h later, reduced fear freezing in context associated with foot-shocks and reduced spatial working memory. Also, social interaction was reduced and immobility time increased in the forced swim test in the mice harboring downregulated mPFC Piccolo and subjected earlier to social-defeat stress. Acute treatment with the antipsychotic risperidone reversed the novelty-induced hyperactivity and disrupted prepulse inhibition of these mice. Novel object recognition, fear-freezing in context or spatial working memory defects were not ameliorated by risperidone, but it should be noted that risperidone appeared to worsen these performances also in the control mice. Based on these results the authors conclude that the downregulated Piccolo in the mouse mPFC models positive, negative and cognitive symptoms of schizophrenia.

  • The results are of interest, experiments appear to be properly performed and the conclusions drawn from the results appear reasonable. However, the authors should themselves carefully once more proof read the whole manuscript and correct inconsistencies.

 I have several concerns regarding the manuscript. These should be carefully addressed.

We appreciate your understanding our study and deeply apologize our lack of explanations and careless mistakes. We responded to your comment point by point.

  • Firstly, it was not clear in what type of neurons Piccolo protein was downregulated in the mPFC. What is known about virally mediated miRNA, which neurons are targeted, how widely distributed? This should be covered in more detail. Secondly, if this protein is presynaptic, should it be downregulated also in the projection sites such as the dorsal striatum and many other well-known projection sites of mPFC projection neurons. Were protein levels analyzed in other brain regions, e.g. striatum, in addition to the mPFC?

For the first question related to the Piccolo downregulation, Pclo miRNA, encoded under CMV promotor, is carried by AAV1, which infects mainly neurons and rarely glial cells. Thus, Piccolo expression was non-specifically suppressed in the neurons in the broad layers of mPFC as shown in Fig. 1A. miR-155 is used as a backbone. The substantial information for targeting is the sequence shown in the method (line 125-127 in the revised manuscript) The comment was added in the revised manuscript (line 124-129). For the second question, it is surely meaningful to confirm the downregulation of Piccolo in the projecting sites. However, there is no antibodies applicable for Piccolo detection in immunohistochemistry in spite of our lots of trials for detection. It is probably difficult to detect by WB because the Piccolo in the terminals of the projecting neurons from mPFC is originally a small portion of the total regional expression.

  • Optogenetic methodology was very shortly described. Please, describe in detail the virus construct (source if commercially obtained). Was the light-cable cannulation (give details) inserted at the same time and together with the microdialysis probe, was this performed at the same time as virus injections? In what type of cells and at what subcellular locations are the virally mediated light-sensitive protein expected to be expressed? What is this protein? How was its expression verified in the neurites projected from the mPFC to the dorsal striatum? In Fig. 3C, after which virus is this GFP expression shown in the dorsal striatum? Comparing this representative image 3C with Fig.1A, why any projections are not observed in Fig.1A image? In Fig. 1A, the GFP expression appears very locally restricted. Could you provide images in both cases where both mPFC and striatum are seen simultaneously? Is the virus construct containing GFP the same in Fig. 1A and 3C?

ChIEF is encoded under human synapsin promotor and carried by AAV9 vectors. Four weeks after the microinjection, guide cannula for an optical fibre and in vivo microdialysis probe was implanted in the dSTR (+ 1.2 mm anterior, 1.0 mm lateral from the bregma and 2.7 mm ventral from the skull). ChIEF selectively expressed in neurons by the effect of the promoter. ChIEF is a light-sensitive protein that generated by introducing some mutagenesis to the original channelrhodopsin 2 (ChR2), and its activation in response to the light resembles more natural spiking patterns than ChR2. The functional expression of ChIEF in the dSTR is confirmed by the response to the optical stimulation. The comment and reference were added in the revised manuscript (line 195-203 and ref [37]). GFP expression shown in Fig. 3C was derived from AAV-EGFP vector used as a control of AAV-miPiccolo injected mPFC to visualize the projection from the mPFC to the dSTR. In the injection site, neurons are majorly infected by virus vector expressing EGFP. Thus, it is difficult to identify the fibers from such neurons because of the occupation by cell body expressing EGFP. It is difficult to prepare the slice with the injection site and the dSTR. We think it is sufficient to conclude that neurons expressing EGFP with the suppression of Piccolo are projected to the dSTR.

  • Title is somewhat misleading, in my opinion, it is not shown in this manuscript that the neuronal disconnection causes the observed behavioral impairments. Also, the abbreviated title is not appropriate (observed impairments may resemble some aspects of schizophrenia, but not convincingly “psychosis”). Please rephrase both titles.

We appreciate your comment. The title was revised as “Schizophrenia-like Behavioral Impairments in Mice with Suppressed Expression of Piccolo in the Medial Prefrontal Cortex.”.  The abbreviated title is not required in this journal, therefore, abbreviated title was deleted.

  • Results shown in Fig. 1 and 2 are not mentioned in the abstract at all.

The description related to Fig. 1 and 2 was added to the abstract in the revised manuscript (line 35-37).

  • Representative images of parvalbumin immunohistochemistry (Fig. 1F) appear to demonstrate that there are even more positive cells in the miPiccolo image. Indicate which cells are counted as positive in these images. Give a micrometer scale for these images and detailed location within the mPFC.

We are deeply sorry that we removed Fig. 1F because we currently cannot access the image of parvalbumin-positive neurons with scale bar. As you pointed out, it is difficult to understand and discuss these images. Fig. 1F and related method were removed.

  • Discussion contains unnecessary repetition. It is recommended that the authors rewrite the discussion to make it more compact.

The first and second paragraphs of the original manuscript were combined in one paragraph. The comment on SNAP-25 in the fifth paragraph in the original manuscript was deleted. The sixth paragraph was also completely deleted.

Other comments

  • Line 66, unusualness. Please change this.

The word “unusualness” was replaced to “impairment” in the revised manuscript (line 66).

  • Line 75, damages, would “insults” fit better here?

The word “damages” was replaced to “insults” in the revised manuscript (line 75).

  • Line 82: given references do not directly link Piccolo to psychostimulant-induced psychosis, please specify this

The reference [19] in the original version is describing the role of Piccolo in the cognitive function and synaptic plasticity. The reference [20] in the original version is showing the protective role of Piccolo against methamphetamine-induced psychosis. The reference [21] in the original version is reporting the relevance between the single nucleotide polymorphism of PCLO and drug dependence-related parameters, which has a small gap from the link Piccolo to psychostimulant-induced psychosis, thus [21] was removed in this context in the revised manuscript (line 83).

  • Line 110, in fear-conditioning and novel object recognition tests, did you administer risperidone on both days, please specify this.

Risperidone was administered 30 min before the fear-conditioning and novel object recognition tests on each day. We added the explanation in the revised manuscript (line 118-119, 841-842, 844-845).

  • Line 216: discrimination of novel object

The phrase “Discrimination of spatial novelty” was replaced to “Discrimination of novel object” in the revised manuscript (line 225).

  • Lines 223-224: how many foot-shocks given during which time, was freezing analyzed immediately after, did it differ?

In the training phase of the fear conditioning test, electrical foot shocks (0.6 mA, five seconds) with an auditory cue were automatically delivered four times with 15 second-intervals. Freezing response were observed 24 hours after the training. The comment was added in the revised manuscript (line 232-234).

  • Lines 128-132: Please, list all behavioral tests here. Spatial working test mentioned two times, why? If pcr, wester blot and immunohistochemistry were performed after behavioral experiments, how did you control possible confounding effects of behavioral tests e.g. fear-conditioning, risperidone treatments, social-defeat, forced-swim test?

Locomotor test, Y-maze test, Novel object recognition test, fear-conditioning test, and PPI test were conducted in this order in the same group of mice.  Spatial context learning was monitored by fear-conditioning test, and spatial working memory was measured in Y-maze task.  These tests for special memory were consisted in two types of tasks.  For mild stress defeat stress exposure, social interaction test, and forced swimming test, another lot of mice was prepared. The mice for PCR, WB and immunohistochemistry did not undergo behavioral experiments.

  • Lines 148-153: What was detection method for parvalbumin immunohistochemistry, did you use perfused and fixed brains, section thickness and plane; sagittal, coronal or horizontal? Give more details about quantification, which parts of mPFC included?

We are deeply sorry that we removed Fig. 1F because we currently cannot access the images of parvalbumin-positive neurons with scale bar.  As you pointed out, it is difficult to understand and discuss these images. Fig. 1F and related method were removed.

  • Lines 309-310: p=0.8 but in Fig.3B left panel marked as significant.

The main text is correct. The symbol for the significance was deleted in the left panel of Figure 3B.

  • Lines 324-326: injected where, what kind of protein is CHIEF? Where is it expressed? Any reference?

ChIEF is a light-sensitive protein that generated by introducing some mutagenesis to the original Channelrhodopsin 2 (ChR2), and its activation in response to the light resembles more natural spiking patterns than ChR2. AAV-ChIEF vectors were microinjected into the mPFC (+ 1.7 mm anterior, 0.3 mm lateral from the bregma, and + 1.5 mm ventral from the skull) as described in the line191-192 in the original manuscript. These related description and reference were added in the revised manuscript (line 193-201 and ref [37])

  • Line 331: If this photosensory protein is expressed in the mPFC, how does optical stimulation in the striatum leads to stimulation?

ChIEF expression in the dSTR is estimated by the GFP image in Fig. 3C. ChIEF was activated by the light by the implanted optical fiber in the dSTR (+ 1.2 mm anterior, 1.0 mm lateral from the bregma and 2.7 mm ventral from the skull). For optogenetic stimulation, the mice received blue light pulses (pulse width, 15 msec; frequency, 10 Hz; intensity, 5 mW, laser, 473 nm) for 15 min. The comment was added in the revised manuscript (line 196-203).

  • Line 339: antagonist for AMPA type of glutamate receptors

The phrase “an antagonist for glutamate receptor” was replaced to “an antagonist for AMPA type of glutamate receptors” in the revised manuscript (line 348).

  • Line 363: miPiccolo enhanced depolarization evoked DA levels in the striatum (Fig. 3G)

The phrase “these responses were suppressed in miPiccolo mice” was replaced to “these responses were enhanced in miPiccolo mice” in the revised manuscript (line 373).

  • Line 377-379 and all other results with two-way ANOVA: Indicate always also the factor (treatment, drug, interaction) when giving the results of two-way ANOVA. Now, it is said that there is a significant increase…, but F and p=0.134?

All the statistical analyses related to the behavioral test was re-performed by one-way ANOVA and post hoc multiple comparisons using Tukey’s test. In the PPI test, the extreme outliers determined by Smirnov-grubbs test were removed. The description was added in the method section of the revised manuscript (line 217-218, 272-279, 389-390, 393-394, 402-403, 406-407, 410-411, 415-420, 431-432, 434-435, 836, 839-840, 842-843, 845-848).

  • Line 396: Did you analyze freezing immediately after foot-shocks, did it differ? Did you analyze cue-induced freezing in a different context? Did it differ?

As described in the method in line 232-234 in revised manuscript, cue-induced freezing behavior was analyzed 24 hours after the training.

  • Line 403: p=0.222? Line 409: p=0.134
  • Lines 423 and 427: give F and p values for each factor (treatment, stress) and interaction

The data related to these were re-analyzed by one-way ANOVA and post hoc multiple comparisons using Tukey’s test.

  • Line 449: Are there glutamatergic interneurons in the mPFC? How downregulated Piccolo affects glutamatergic pyramidal neurons sending collaterals to other pyramidal neurons and GABAergic interneurons? What is known about Piccolo in cortical GABAergic interneurons?

Glutamatergic interneurons in the mPFC expresses NMDA-type and AMPA-type receptors, whose impairment is related to schizophrenia (Reviewed in “Cell type-specific development of NMDA receptors in the interneurons of rat prefrontal cortex” Neuropsychopharmacology 2009, 34, 2028-2040.). Piccolo is physiologically located in the terminals of glutamatergic and GABAergic neurons. Thus, Piccolo downregulation reduce the activity of neuronal transmission. The related comments were added in the revised manuscript (line 453).

  • Line 469: medium spiny GABAergic neurons?

The phrase “provide excitatory input to mediate GABA neurons” was replaced to “provide excitatory input to GABA neurons” in the revised manuscript (line 482-483).

  • Line 494: remove some

“some” was removed

  • Line 521: disconnection

The phrase “disconnect” was replaced to “disconnection” in the revised manuscript (line 534).

  • Lines 528-536: unnecessary repetition

The paragraph you pointed out was removed.

  • Line 533: In Fig 4B, you show that risperidone improved PPI in miPiccolo at least at 82dB prepulse, did two-way ANOVA support this observation? Please, add two-way ANOVA results and comment of this in Results section.

The data were re-analyzed by one-way ANOVA and post hoc multiple comparisons using Tukey’s test.

  • Lines 539-541: What is the policy of this journal in reporting unpublished findings?

We showed such data in the supplementary figure 2 (line 543, 545).

  • Line 542: Is your method knocking down gene or its expression?

We knocked down Pclo gene by microRNA expression.

  • 1D: check y-axis

The y-axis “Expression of Protein (Relative to GAPDH)” was replaced to “Expression of Protein (Relative to alpha-Tubulin)” in Fig 1D. Also typos were corrected “alpha-Tublin” to “alpha-tubulin” beside the WB in Fig 1D and the revised manuscript (line 162, 792).

  • 2: figure legend, please give plane of slices used

We added some explanation was added in the revised manuscript (line 798-799).

  • 4: figure legend, please mention when risperidone was given in the fear conditioning and object recognition tests, 1st or 2nd day?

We added the descriptions in the revised manuscript (line 841-842, 844-845).

  • 5B: check y-axis

The y-axis “Interaction time (sec)” was replaced to “Immobility time (sec)” in Fig 5B.
